# First Isolation and Characteristics of Bovine Parainfluenza Virus Type 3 from Yaks

**DOI:** 10.3390/pathogens11090962

**Published:** 2022-08-24

**Authors:** Yunxin Ren, Xi Chen, Cheng Tang, Hua Yue

**Affiliations:** College of Animal and Veterinary Sciences, Southwest Minzu University, Chengdu 610041, China

**Keywords:** bovine parainfluenza virus type 3, yak, isolation, genome characteristics

## Abstract

The yaks belong to the genus *Bos* within the family *Bovidae* that live in the Tibet Plateau and is an indispensable economic resource for the local herders. Respiratory tract infections are common diseases in yaks caused by various pathogens; however, there have been no reports of bovine parainfluenza virus type 3 (BPIV3) infection. This study was conducted to investigate the pathogens and analyze their characteristics from the four yak lung samples with severe respiratory tract infection symptoms in the yak farm. Results showed that out of four lung samples, three were identified as BPIV3-positive by RT-PCR. A BPIV3 strain (10^6.5^ TCID_50_/mL) was successfully isolated from the BPIV3-positive lung samples using Madin–Darby bovine kidney cells. The isolate caused systemic infection in the BALB/c mice and induced pathological changes in the lungs. Moreover, three complete BPIV3 genomes were amplified from the clinical samples. Phylogenetic trees based on the complete genomes, hemagglutinin-neuraminidase protein (HN), phosphoprotein (P), and large polymerase subunit protein (L) amino acid sequences showed that the complete BPIV3 genomes belonged to BPIV3 genotype C, and clustered into a large branch with the Chinese strains, although the three yak BPIV3 strains were clustered into a small branch. Compared to known BPIV3 genotype C strains in GenBank, the three genomes of yak BPIV3 showed four identical amino acid mutations in the HN, P and L proteins, suggesting a unique genetic evolution of BPIV3 in yaks. This study first isolated and characterized the BPIV3 from yaks, which contributed to the understanding of the infection and evolution of BPIV3 in yaks in the Tibet Plateau.

## 1. Introduction

Bovine respiratory disease complex (BRDC) is a multi-pathogenic syndrome, which is a major health threat to bovines. Bovine parainfluenza virus type 3 (BPIV3), as the primary causative agent of BRDC, has caused severe economic losses to the cattle industry worldwide [1,2]. In addition, BPIV3 infections have been found in a wide variety of mammals, including pigs [3], camels [4], dolphins [5], sheep [6], etc., and the molecular mechanism of cross-species transmission of BPIV3 is still undetermined.

BPIV3 is an enveloped, non-segmented, negative-strand RNA virus that belongs to the family *Paramyxoviridae* [7]. The genome length is 15434–15504 bases, encoding six structural proteins, named nucleocapsid protein (N), phosphoprotein (P), large polymerase protein (L), matrix protein (M), fusion protein (F), hemagglutinin-neuraminidase protein (HN), and three non-structural proteins (C, D, and V) [8,9,10,11,12]. Based on phylogenetic analysis, BPIV3 is classified into three genotypes [13,14]. Genotype A has been detected in the United States, China, Argentina, and Japan [13,14,15,16]. Genotype B has been detected in Australia and Argentina [15,16], and genotype C has been detected in China, South Korea, the United States, Japan, and Turkey [2,14,15,17,18,19]. Until now, only genotype A and C were found in China [14,20], and the BPIV3 genotype C is mainly prevalent in the Inner Mongolia province, and the results of seroepidemiological investigation show that BPIV3 has been widely distributed in China [21], whereas the molecular epidemiological investigations of BPIV3 are limited.

The yak (*Bos grunniens*) is a distinct species within the family *Bovidae* that inhabits the high-altitude Tibet Plateau of China, Nepal, India, Pakistan, Kyrgyzstan, Mongolia, and Russia [22]. Yaks are essential sources for the local herdsmen [23]. Respiratory diseases are common in yaks and cause significant economic losses to the yak industry [24], and BRDC-related viruses have been identified in the yaks, including bovine coronavirus (BCoV) and bovine viral diarrhea virus (BVDV) [25,26]; however, information regarding the BPIV3 infection in yaks is largely unknown. The purpose of this study was to investigate the pathogens causing BRDC in a yak farm and analyze its characteristics.

## 2. Results

### 2.1. Pathogens Detection and Isolation

Four lung tissue samples of yaks were collected for pathogen detection, and four samples tested as BPIV3-positive by RT-PCR assay; BCoV, BVDV, bovine herpesvirus 1 (BHV-1), bovine respiratory syncytial virus (BRSV), *Mycoplasma bovis* (*M. bovis*), *Pasteurella multocida* (*P. multocida*), and *Mannheimia haemolytica* (*M. haemolytica*) were not detected in lung samples of yaks.

After three generations of blind passaging on Madin–Darby bovine kidney (MDBK) cells, one sample showed evident cytopathic effect (CPE) at 72 h. The CPE was characterized by cell round shrinkage, shedding, and fusion (Figure 1A), and no CPE was observed in the negative control (Figure 1B). Stable CPE appeared in the fourth to sixth passages, and the cultures of the sixth passage were used for virus purification. The virus titer was 10^6.5^ TCID_50_/mL. The other two samples of cell cultures showed no CPE.

### 2.2. Indirect Immunofluorescence Assay (IIFA), Transmission Electron Microscopy, and Hemagglutination (HA) Test

IIFA results showed that virus particles were in the cytoplasm (Figure 2A). Transmission electron microscopy showed that the virus particle is pleomorphic, frequently spherical, the diameter of the virus was about 200 nm, with envelope and filamentous structures outside the virus (Figure 2B), and the structure of the virus was similar to those of the paramyxovirus group [27,28,29]. The HA test showed the isolated agglutinated chicken red blood cells at 4 °C; the blood coagulation titer was 1:16. The BPIV3 isolate from the yak was named Yak/GZ01/20/CH. 

### 2.3. Experimental Infection of Yak/GZ01/20/CH to BALB/c Mice

The infected mice showed clinical manifestations of decreased appetite, rough hair coat, and depression at 2–9 days post-inoculation (DPI) and returned to normal clinical manifestations at 10 DPI. The real-time RT-PCR assay was used to detect the distribution of the virus in infected BALB/c mice. The isolate could be detected in blood and tissues (Table 1). Gross lesions characterized by hyperemia and consolidation were observed in the lungs of infected mice. Histopathological changes were present in infected mice at 3 DPI, including alveolar septal thickening, lymphocyte and macrophage infiltration, fibrous hyperplasia, and serous exudation (Figure 3A,B). Histopathological injuries were alleviated at 15 DPI (Figure 3C,D).

### 2.4. Genomic Characteristics of BPIV3 from Yak

Three complete genomes of the yak BPIV3 strains (containing isolate Yak/GZ01/20/CH) were obtained from the clinical samples and were named Yak/LT01/20/CH (GenBank numbers OM782290), Yak/LT02/20/CH (GenBank numbers OM782291), and Yak/GZ01/20/CH (GenBank numbers OM621819), respectively. The length of the three complete genomes was 15,474 nt and the G + C contents were 36.7%, 36.8%, and 36.9%, respectively. The phylogenetic tree, based on all the complete genomes, HN, P, and L amino acid (aa) sequences from GenBank, showed that the three strains belonged to BPIV3 genotype C and clustered into a large branch with the Chinese strains (NX49, XJA13, and SD0835); however, the three yak BPIV3 strains were clustered into a small branch (Figure 4). Further analysis revealed that the three complete genomes of the yak BPIV3 strains shared 99.96–99.98% nucleotides (nt) and 99.89–99.96% aa identity with one another and shared 99.6–99.8% nt identity and 84.4–99.7% aa identity with 13 known complete genomes of BPIV3 genotype C strains from GenBank. Interestingly, compared to 13 known genomes of BPIV3 genotype C strains, the three yak BPIV3 genotype C strains shared identical nt substitutions, which resulted in aa mutations in the P, HN, and L protein (Table 2). In the analysis of recombination, there were no recombination events observed in yak BPIV3 complete genomes.

The length of the three HN, P, and L sequences of the yak BPIV3 genotype C strains were 1719 nt (encoding 572 aa), 1803 nt (encoding 600 aa), and 6702 nt (encoding 2233aa), respectively. The three HN sequences of the yak BPIV3 strains shared 100% nt and 100% aa identity with one another and shared 79.5–99.6% nt identity and 82.9–99.1% aa identity with all 114 complete HN sequences of BPIV3 strains from GenBank. Sequence analysis showed that the three HN sequences shared two identical aa mutations; one was I49V, and the other was I550M, which was also found in the Japanese strain NM2. The three P sequences of the yak BPIV3 strains shared 100% nt and 100% aa identity with one another and shared 79.0–99.7% nt identity and 69.6–99.5% aa identity with all 102 complete P sequences of BPIV3 strains from GenBank. Sequence analysis showed that the three P sequences shared an identical unique aa mutation (Y/N261H) compared to all the BPIV3 genotype C strains. Interestingly, compared with all the P sequences of the BPIV3 genotype A, B, and C strains, the aa mutation was also found in the three BPIV3 genotype A strains (Chinese strain NM09, Egyptian strain 3/Egypt/2014, and American strain TVMDL60); the three L sequences of the yak BPIV3 strains shared 99.98–99.99% nt identity and 99.91–99.96% aa identity with one another and shared 83.3–99.8% nt identity and 91.8–99.7% aa identity with 27 known complete L sequences of BPIV3 strains. Notably, sequence analysis showed that the three L sequences from yak shared an identical aa mutation (S1098N) compared to the BPIV3 genotype C strains. Compared with all the L sequences of the BPIV3 genotype A, B, and C strains, the aa mutation was also found in the ten BPIV3 genotype A strains and all four BPIV3 genotype B strains.

## 3. Discussion

BRDC is one of the most significant diseases of cattle, which has caused severe economic losses to the cattle industry worldwide [1]. Many pathogens can lead to the emergence of BRDC, among which BPIV3 is one of the important pathogens [2]; however, the information on BPIV3 infection in yak is still unknown so far. In this study, three of the four lung tissue samples of yak were detected as BPIV3-positive by RT-PCR, and BPIV3 was successfully isolated from one of the three samples; the unsuccessful virus isolation from the then-remaining-symptomatic yaks could be related to the absence of infectious virus particles or the low viral load of sampling. Despite respiratory diseases being common in yaks, relatively little is known of BRDC-related pathogens [24]. The result of this study was the first time to demonstrate the presence of BPIV3 in yaks, which is of great significance for the diagnosis and prevention of yak BRDC. Moreover, previous reports have ensured that the BALB/c mice infection model could cast light on the genotype C of the BPIV3 infection process and pathogenesis [30], and the genotype C isolate can cause systemic infection and gross lesions in lungs of BALB/c mice, which is consistent with the previous conclusion of BPIV3 infection in calves and BALB/c mice [30,31], indicating that the BALB/c mice model may be used to evaluate the vaccine efficacy. Further epidemiological investigations of BPIV3 in yaks are necessary.

Phylogenetic analyses indicated that the yak BPIV3 were closely related to the Chinese BPIV3 genotype C strains, whereas the BPIV3 strains in yaks have unique evolutionary characteristics. Based on these results, and the frequent local animal trade, we hypothesized that yak BPIV3 might have been transmitted from cattle. Interestingly, compared to all the complete HN, P, and L sequences of BPIV3 from GenBank, four unique aa mutations were found in the HN (I49V, I550M), P (Y/N261H), and L (S1098N) proteins; however, the biological significance is unclear. Paramyxoviruses are similar in genome structure and replication strategy, and the HN, P, and L proteins were closely related to viral virulence, viral transcription and replication, as well as the production of interferons (IFNs), respectively [32,33,34]. Studies on other paramyxoviruses have proved that single aa mutations on HN, P, and L proteins have important biological significance. For instance: Studies on the Newcastle disease virus in the family *Paramyxoviridae* have shown that a single aa mutation in HN significantly altered the viral pathogenicity [35]; studies on the human parainfluenza virus type 3 have shown that a single aa mutation in L can regulate the levels of virus gene expression [36]. Therefore, the specific effects of the aa mutations in HN, P, and L proteins on the biological characteristics of the yak BPIV3 need further investigation. Moreover, information on the genetic evolution of BPIV3 in different hosts is limited. Several studies on other paramyxoviruses showed that the viral genome undergoes adaptive mutations in different host cells. For example, the Sendai virus has a single aa mutation in the P and L proteins, respectively, by changing the host cells [37]. Therefore, whether these unique aa mutations of BPIV3 strains were host-adaptive mutations in the yaks that live in the unique natural environment of the Tibet Plateau (i.e., average altitude >4000 m, hypobaric hypoxia, and low temperature) [38] needs to be determined.

In conclusion, we found for the first time that BPIV3 can infect yaks and we successfully isolated BPIV3 from yaks; the isolate caused systemic infection in the BALB/c mice and induced pathological changes in the lungs. Three complete genomes were successfully obtained and analyzed, which suggested the unique evolutionary trends of the yak BPIV3. All these results can contribute to a better understanding of the infection and genetic evolution of BPIV3.

## 4. Materials and Methods

### 4.1. Clinical Samples and Cell Lines

In October 2020, a severe respiratory disease broke out on a yak farm in Litang County, Sichuan Province. The symptoms of sick yaks included coughing, runny nose, shortness of breath, and fever. Lungs were collected from four ill yaks (five months old) and sent to our laboratory for diagnosis. All samples were stored at −80 °C. MDBK cells were used for virus isolation.

### 4.2. Sample Processing and Nucleic Acid Extraction

The lung tissue samples were ground with liquid nitrogen, and phosphate-buffered saline (PBS; 1:5) was added. The samples were lysed by three times repeated freezing–thawing, and then the supernatants were collected after centrifugation at 10,000 r/min for 10 min. The Viral DNA Kit (Omega Bio, New York, NY, USA) was used to extract viral DNA; viral RNA was extracted using RNAios Plus (TaKaRa Bio, Inc., Kusatsu, Japan) according to the manufacturer’s instructions. cDNA was synthesized using a commercially available PrimeScript™ RT reagent kit (TaKaRa Bio Inc.) and stored at −20 °C.

### 4.3. Pathogens Detection

The common BRDC-related pathogens were detected using a specific RT-PCR assay as previously reported [14,21,24,39,40,41], including BPIV3, BCoV, BVDV, BHV-1, BRSV, *M. bovis*, *P. multocida* and *M. haemolytica*. All primer sequences were listed in Table 3, and each reaction was performed in 25 μL reaction volume with 0.05 μM forward primer, 0.05 μM reverse primer, template 2 μL, 12.5 μL PCR Master Mix (2 × Premix), and an appropriate volume of double-distilled water. Then, the amplification products were analyzed by 1.5% agarose gel electrophoresis.

### 4.4. Virus Isolation, Purification, and Titer Determination

The positive BPIV3 samples detected using the RT-PCR assay were centrifuged at 10,000 r/min for 10 min at 4 °C, and the supernatants were collected and filtered through 0.22 µm filters. The filtered supernatants were inoculated into MDBK cells maintained in 2% fetal bovine serum (FBS) (Thermo Fisher, Massachusetts, Boston, USA) and cultured at 37 °C in 5% CO_2_; then, cell cultures were observed daily for the forming of the CPE. The isolates were considered negative if no CPE appeared after three blind passages. The cell cultures in which CPE appeared were used for the virus purification and titration assay after the time CPE was stable. The virus was purified via plaque purification (three times) [42], and the viral titer (TCID_50_/mL) was determined in MDBK cells according to the Reed–Muench method [43].

### 4.5. IIFA

The IIFA was conducted to detect the specific antigens in MDBK cells. The low-density monolayer of cells was incubated with aliquots of 0.1 multiplicity of infection (MOI) of viruses at 37 °C for 1 h, then replaced with Dulbeccos Modified Eagle Medium (DMEM) (Thermo Fisher, Massachusetts, Boston, MA, USA) containing 1% FBS. After 24 h, the media was removed from each well and fixed in cold acetone (−20 °C) for 30 min. After three washes with PBS (pH 7.2), bovine serum albumin was diluted in PBS to a final concentration of 3%, and then the cells were blocked for 1 h. Cells were incubated with a primary rabbit anti-BPIV3 whole-virus polyclonal antibody developed by our laboratory at 1:200 for 30 min at 37 °C. Finally, positive cells were visualized using fluorescein isothiocyanate-conjugated anti-rabbit IgG (1:1000) for 2 h at 37 °C. Cell nuclei were stained with 4’,6-diamidino- 2-phenylindole (1 mg/mL, 1:2000 in PBS) for 10 min. Fluorescence signals were observed using fluorescence microscopy.

### 4.6. Transmission Electron Microscopy

The yak BPIV3 strain was inoculated into MDBK cells. Once the CPEs of 80–90% were developed, cells were frozen–thawed three times and centrifuged at 10,000 r/min at 4 °C for 10 min to remove cellular debris. Supernatant were stained with 2% phosphotungstic acid for 2 min and observed under a transmission electron microscope.

### 4.7. HA Test

The HA titer was measured in pH 7.2 of PBS at 4 °C to determine the HA activities of the yak BPIV3 strain. Tests were carried out in 96-well V-plates (25 µL of sample dilution per well). Two-fold serial dilutions of the viral supernatant were made in PBS. Then, equal amounts of a suspension containing 1% chicken erythrocytes were added to each well. Results were read after 1 h at 4 °C, and HA activity was evaluated by the appearance or absence of a red cell button. HA titers were calculated as the reciprocal of the highest virus dilution showing complete HA.

### 4.8. Experimental Infection of BALB/c Mice with the BPIV3 Isolate from Yak

Twenty-four specific pathogen-free BALB/c mice were randomly divided into an infected group (18 mice) and a control group (6 mice). Under ether anesthesia, the infected group was inoculated with 100 μL of the virus (10^6.5^ TCID_50_/mL) in the nasal cavity, and the uninfected group was inoculated with supernatants of uninfected MDBK cells. The clinical manifestations were recorded daily. Three mice in the infected group and one control mouse were euthanized at varying intervals (1, 3, 6, 9, 12, and 15 DPI). Heart, liver, spleen, lung, kidney, trachea, and blood samples were collected from each mouse, and the samples were immediately frozen and stored at −80 °C to detect tissue virus distribution using real-time RT-PCR [44]. The tissue samples were fixed in 10% neutral buffered formalin for a histopathology examination.

### 4.9. Complete Genome Amplification

Based on known BPIV3 genotype C genomes (GenBank numbers KT071671), eight pairs of primers were designed for amplifying the genome sequence of BPIV3 strains from three positive clinical samples. Primer sequences are shown in Appendix A. PCR products were purified and cloned into the pClone007-T simple vector (Tsingke Bio, Inc.) for sequencing by Tsingke Biotechnology Co. (Chengdu, China).

### 4.10. Sequence and Phylogenetic Analysis

The gene sequences were assembled using SeqMan software (Version 7.0; DNA Star, Madison, WI, USA). Multiple sequence alignment was performed with the MEGA 7.0 software to build a neighbor-joining phylogenetic tree with 1000 bootstrap support [45], and BPIV3 genotypes were determined based on complete genome phylogenetic analysis. Recombination events were assessed using Recombination Detection Program 4.0 (RDP 4.0, version 4.96) [46].

## Figures and Tables

**Figure 1 pathogens-11-00962-f001:**
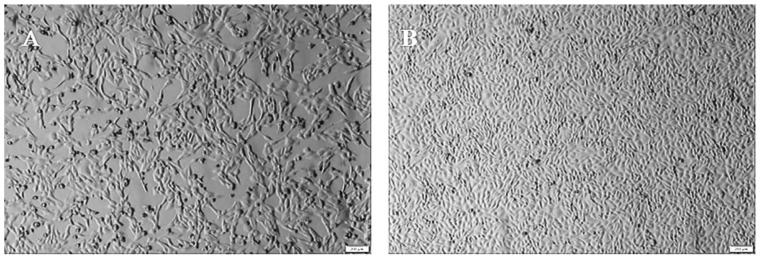
The CPE of isolate in Madin–Darby bovine kidney (MDBK) cells. (**A**) CPE at 72 hours after virus inoculation (shrinkage, shedding, and fusion). (**B**) Control cells after 72 h of culture.

**Figure 2 pathogens-11-00962-f002:**
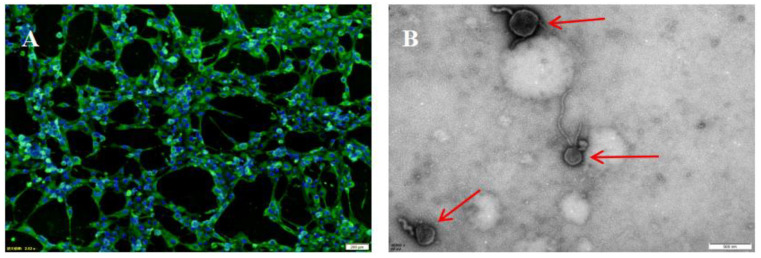
Microscopic examination of BPIV3 isolated from yaks. (**A**) The immunofluorescence test of isolate (40×). (**B**) Transmission electron micrograph of the isolates in MDBK cell cultures, and the red arrow points to the virus particle (×40,000).

**Figure 3 pathogens-11-00962-f003:**
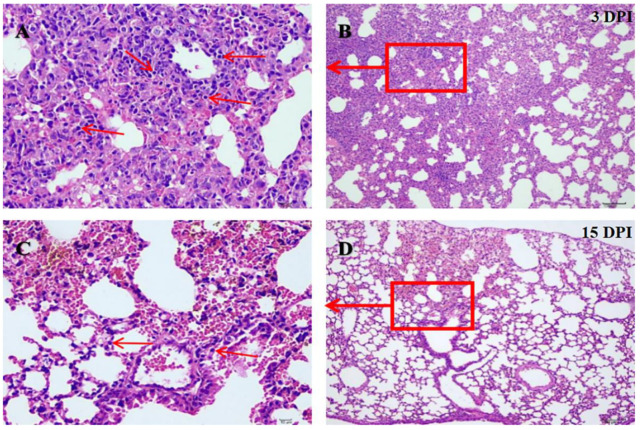
The lung sections of infected mice after hematoxylin and eosin (H & E) staining. (**A**) At 3 DPI, alveolar septal thickening, lymphocyte and macrophage infiltration, fibrous hyperplasia, and serous exudation were found in lung sections of mice in the infected group. The red arrow points to the lymphocyte infiltration (×400). (**B**) The red boxes indicate the magnified viewing area in A figure (×100). (**C**) At 15 DPI, lung sections of mice in the infection group showed a small amount of inflammatory cell infiltration. The red arrow points to the lymphocyte (×400). (**D**) The red boxes indicate the magnified viewing area in C figure (×100).

**Figure 4 pathogens-11-00962-f004:**
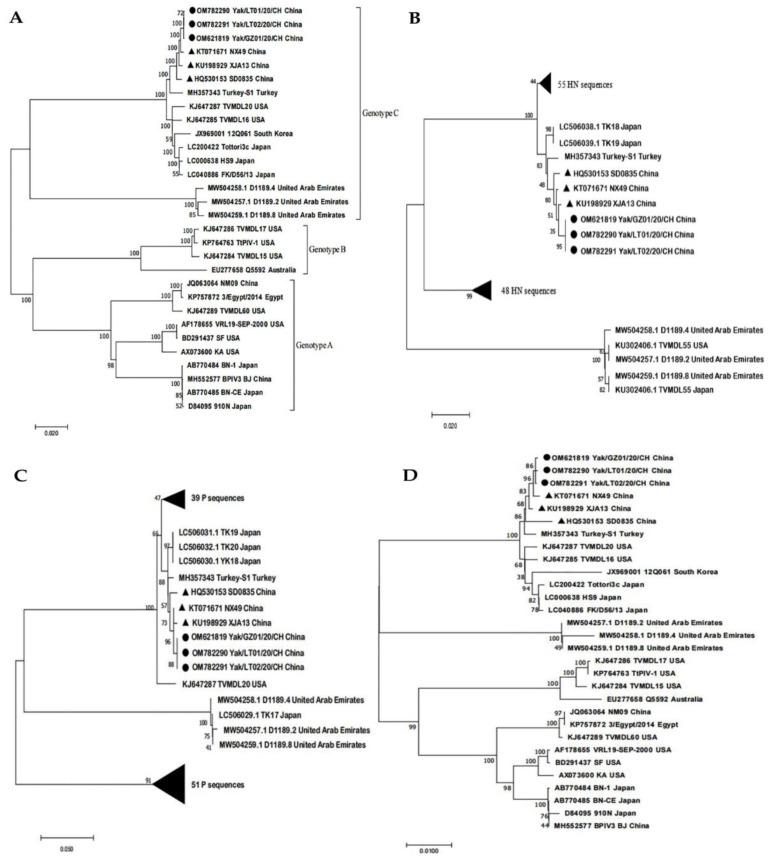
Phylogenetic analysis of BPIV3 strains based on all the complete genome sequences (**A**), all the amino acid sequences of the HN (**B**), P (**C**), and L (**D**). Sequences were compared by MEGA 7.0 software; neighbor-joining method was used to construct phylogenetic trees, and bootstrap values of 1000 replicates were calculated. ◄ represents the HN and P sequences and the number of sequences not shown, respectively, ● represents the yak BPIV3 sequences from this study, ▲ represents the other BPIV3 genotype C sequences in China.

**Table 1 pathogens-11-00962-t001:** Tissue distributions of BPIV3 isolate Yak/GZ01/20/CH in infected mice.

Organs	Day 1			Day 3			Day 6			Day 9			Day 12			Day 15		
	1	2	3	4	5	6	7	8	9	10	11	12	13	14	15	16	17	18
Lung	−	+	+	+	+	+	+	+	−	+	+	+	−	+	−	+	+	−
Trachea	+	+	+	+	+	+	+	+	+	−	+	−	+	+	+	−	−	+
Heart	−	−	−	−	+	−	−	−	+	+	−	+	+	−	−	−	−	−
Liver	−	−	−	+	+	+	−	+	+	+	+	−	−	−	−	+	−	−
Spleen	−	−	−	−	+	−	+	+	−	+	+	−	−	−	+	−	−	−
Kidney	−	−	−	+	−	+	+	−	−	−	+	+	+	−	−	−	−	−
Blood	−	−	−	+	+	−	+	−	−	+	+	−	−	+	−	+	−	−

Note: +: BPIV3 positive; −: BPIV3 negative; 1–18: Mice of the experimental group.

**Table 2 pathogens-11-00962-t002:** Nucleotide and amino acid differences of three complete genomes of yak BPIV3 strains compared with the 13 complete genomes of BPIV3 genotype C strains in GenBank.

Gene Name	Nucleotide Differences			Amino Acid Differences		
	13 BPIV3 Genotype C Strains	Position	Three Yak BPIV3 Strains	13 BPIV3 Genotype C Strains	Position	Three Yak BPIV3 Strains
N	T	1401	C	-	-	-
P	T/A	781	C	Y/N	261	H
M	T	321	C	-	-	-
F	A; C; T; C	81; 714; 1035; 1380	G; T; G; T	-	-	-
HN	G; G	145; 1650	A; A	I; I	49; 550	V; M
L	G; T	3293; 4389	A; C	-; S	-; 1098	-; N

Note: -: nonsense mutation (amino acid level).

**Table 3 pathogens-11-00962-t003:** Primer information.

Pathogens	Primer Name	Sequence (5′–3′)	Size (bp)	Target Gene
BPIV3	BPIV3-F	GAATGACTCATGATAGAGGTAT	647	HN
	BPIV3-R	AGGACAACCAGTTGTATTACAT		
BVDV	BVDV-F	ATGCCCTTAGTAGGACTAG	287	5′ UTR
	BVDV-R	TCAACTCCATGTGCCATGT		
BHV-1	BHV-F	ATGCCGCGATACAACTACACTGAAC	921	gD
	BHV-R	TTATTCGAGGCTCGGCCAGCCTT		
BRSV	BRSV-F	ATGGCTCTTAGCAAGGTC	459	N
	BRSV-R	AGAGTCATGTCTGTATTC		
BCoV	BCoV-F	CTAGTAACCAGGCTGATGTCAATACC	81	N
	BCoV-R	GGCGGAAACCTAGTCGGAATA		
*M. bovis*	M.bovis-F	TAATTTAGAAGCTTTAAATGAGCGC	238	uvrC
	M.bovis-R	CATATCTAGGTCAATTAAGGCTTTG		
*P. multocida*	P-F	ATCCGCTATTTACCCAGTGG	460	KMT1
	P-R	GCTGTAAACGAACTCGCCAC		
*M. haemolytica*	M-F	GCAGGAGGTGATTATTAAAGTGG	206	lktD
	M-R	CAGCAGTTATTGTCATACCTGAAC		

## Data Availability

All the sequencing results in this study have been submitted to the GenBank database under accession numbers OM621819, OM782290 and OM782291.

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
