# Peer review of "First Isolation and Characteristics of Bovine Parainfluenza Virus Type 3 from Yaks"

_pathogens, 2022, doi:10.3390/pathogens11090962_

Round 1
Reviewer 1 Report
Dear Authors,
Please refer to the attached file for comments.

Reviewer 2 Report
Ren and colleagues isolated one BPIV3 from a clinical sample taken from a Yak.
In total, they characterized 3 BPIV3 complete genomes identified in Yaks and compared these genomes with some references from the Genbank. Although purely descriptive the study is interesting because it shows some unique mutations in the BPIV3 strains isolated from a Yak.
The authors need to present a table showing the position of all differences in the genome of BPIV3 identified in Yaks compared with other references.
The phylogenetic analysis must be summarized, it is too repetitive, and also combine all trees in a single figure.
Minor concerns:
Page 2 line 80: 24 is this corrected?
Line 99 (Table 1) Identify the animals used in the experiments with numbers and not letters
Line 109: remove “1.”
Lines 120-123: This sentence is flawed. Rewrite it.
Line 128 (figure 4 legend): change repeats by replicates or replications
Lines 186-189: Rewrite this sentence.
Lines 214-215: The sentence is wrong, please rephrase it clearly.
Lines 220-221: BPIV3 was not isolated firstly in Yaks, rephrase this sentence.
The figure shows that electron microphotography has lots of debris.
You should indicate the lymphocyte infiltration with an arrow in figure 3
Reviewer 3 Report
See attached.

Round 2
Reviewer 1 Report
The revised manuscript has been properly revised.
Reviewer 2 Report
All my comments were fully addressed in the new version of the manuscript.
Reviewer 3 Report
My comments and suggestions were addressed in the updated manuscript.